# Integrins Have Cell-Type-Specific Roles in the Development of Motor Neuron Connectivity

**DOI:** 10.3390/jdb7030017

**Published:** 2019-08-27

**Authors:** Devyn Oliver, Emily Norman, Heather Bates, Rachel Avard, Monika Rettler, Claire Y. Bénard, Michael M. Francis, Michele L. Lemons

**Affiliations:** 1Department of Neurobiology, University of Massachusetts Medical School, Worcester, MA 01605, USA; 2Department of Biological and Physical Sciences, Assumption College, Worcester, MA 01609, USA; 3Department of Biological Sciences, University of Quebec Montreal, Montreal, QC H2L 2C4, Canada

**Keywords:** axon guidance, axon outgrowth, commissure, integrin, *C. elegans*, GABAergic motor neuron, *ina-1*

## Abstract

Formation of the nervous system requires a complex series of events including proper extension and guidance of neuronal axons and dendrites. Here we investigate the requirement for integrins, a class of transmembrane cell adhesion receptors, in regulating these processes across classes of *C. elegans* motor neurons. We show α integrin/*ina-1* is expressed by both GABAergic and cholinergic motor neurons. Despite this, our analysis of hypomorphic *ina-1(gm144)* mutants indicates preferential involvement of α integrin/*ina-1* in GABAergic commissural development, without obvious involvement in cholinergic commissural development. The defects in GABAergic commissures of *ina-1(gm144)* mutants included both premature termination and guidance errors and were reversed by expression of wild type *ina-1* under control of the native *ina-1* promoter. Our results also show that α integrin/*ina-1* is important for proper outgrowth and guidance of commissures from both embryonic and post-embryonic born GABAergic motor neurons, indicating an ongoing requirement for integrin through two phases of GABAergic neuron development. Our findings provide insights into neuron-specific roles for integrin that would not be predicted based solely upon expression analysis.

## 1. Introduction

Assembling neuronal networks requires orchestration of neuronal outgrowth in a spatially and temporally controlled manner. During development, neurons extend axons and dendrites (collectively referred to as neurites) in a tightly regulated manner to establish connections with their synaptic partners. Neurites must extend through diverse extracellular terrains to reach their appropriate partners and form a functional circuit. Impaired neurite extension and guidance can lead to improper neuronal connectivity, potentially contributing to neurodevelopmental disorders.

Integrins are a family of cell surface receptors [1,2,3,4] that are crucial for proper development of the nervous, vascular, and immune systems, as well as epithelial tissues [5,6,7,8,9,10]. There has been longstanding interest in these receptors based on their ability to bind to over 20 extracellular matrix molecules [4,11], and to promote cell adhesion and motility. More recently, integrins have been shown to also interact with axon guidance molecules including netrin [12,13,14], semaphorins [15,16,17], slit [18], and ephrins [19,20,21], as reviewed in [22]. Integrins are a family of heterodimeric receptors consisting of an α and a β subunit. In humans, there are at least 18 α subunits and 8 β subunits, combining to form at least 25 different integrins [4,23]. A single integrin can bind to a wide variety of ligands and many ligands can bind to multiple integrins [4,7,11,24,25]. In addition, multiple integrins are often expressed within a single cell, greatly enhancing the complexity of these receptors, and making it challenging to separate possible effects of redundancy. Much of our understanding of integrins has stemmed from in vitro and model organism studies, the latter of which offer a simplified genetic landscape for in vivo analyses of integrin function in specific tissues, particularly the nervous system.

In particular, the genome of the nematode *Caenorhabditis elegans* encodes only one β subunit (PAT-3) [26], which heterodimerizes with one of two available α subunits, PAT-2 or INA-1, to form only 2 distinct integrins. While a complete loss of integrin signaling causes early embryonic lethality [26,27], we have taken advantage of a hypomorphic mutation in α integrin/*ina-1* (*ina-1(gm144))* [28,29] to more closely examine the role of α integrin/*ina-1* in the neurite patterning of specific neuron classes in vivo, using the development of commissural neurites of *C. elegans* motor neurons as a model.

The cell bodies of cholinergic and GABAergic motor neurons are located in the ventral nerve cord, and a subset extend commissures to the dorsal nerve cord. Using cell-specific expression of fluorescent reporters, we can label informative subsets of motor neurons, including their neurites, to investigate the involvement of integrins in their development.

Here we show that decreased function of *ina-1* disrupts GABAergic (D) motor neuron commissural outgrowth and guidance, consistent with previous studies [28,29,30,31]. In contrast, cholinergic commissures are not significantly affected by mutation of *ina-1*, suggesting a neuron class-specific role of integrin in commissural patterning. GABAergic commissural defects in *ina-1(gm144)* animals can be rescued by expression of wild type *ina-1*. Despite the selective effects of the *ina-1(gm144)* mutation on GABAergic commissural patterning, we observed *ina-1* expression in both cholinergic and GABAergic motor neurons, perhaps pointing towards a higher degree of redundancy in guiding the extension of cholinergic commissures. Lastly, we demonstrate that α integrin/*ina-1* is important for proper guidance of commissures from both embryonic and post-embryonic born GABAergic neurons. Our findings reveal a neuron class-specific role for integrins in establishing motor neuron connectivity during development.

## 2. Materials and Methods

### 2.1. C. elegans Strains

*C. elegans* strains used in this study are listed in Appendix A. All strains were derivatives of the N2 Bristol strain (wild type) and maintained at 20–24 °C on nematode growth media (NGM) plates seeded with *Escherichia coli* strain OP50 as described [32]. Transgenic strains were made by microinjection of plasmid DNA to achieve transformation as described previously [33] and identified with co-injection marker *lgc-11::GFP* (pharyngeal marker) [34]. Integrated lines were produced by X-ray irradiation and outcrossed to wild type.

### 2.2. Construction of Plasmids

As described previously [34,35], plasmids were made using the two-slot Gateway Cloning system (Invitrogen, Carlsbad, CA, USA). Plasmids used in this study are listed in Appendix A, and were confirmed by restriction digest and sequencing.

### 2.3. ina-1 Reporter and Rescue Constructs

*ina-1* cDNA was RT-PCR amplified from *C. elegans* RNA using superscript III and ligated into pDEST-163. Rescuing *ina-1* cDNA constructs were generated by recombining pDEST-202 (*ina-1* cDNA) with pENTR-3′-72 (*ina-1* native promoter), pENTR-3′-*rgef-1* (pan-neuronal-specific promoter), pENTR-3′-*unc-47* (GABA-specific promoter), or pENTR-3′-*unc-17β* (cholinergic-specific promoter) to generate pDO121 (P*ina-1::ina-1cDNA*), pDO81(P*rgef-1::ina-1cDNA)*, pDO117 (P*unc-47::ina-1cDNA)*, and pDO75 (P*unc-17β::ina-1cDNA)*, respectively. The *ina-1* promoter region was amplified from pENTR-5′-*ina-1* that was generated from wild type genomic DNA (4042 bp relative to start). The *ina-1* promoter was cloned into pENTR-72 and was then recombined with pDEST-94 (GFP) to generate pDO119 (P*ina-1*::GFP).

### 2.4. Microscopy

Worms were immobilized with sodium azide (0.3 M) on 2% or 5% agarose pads. Larval stage 4 (L4) hermaphrodites were examined unless otherwise noted. Images were collected using either a 3i (Intelligent Imaging Innovations, Denver, CO, USA) Everest spinning-disk confocal microscope at 0.3 µm/step using a 63x objective or a Zeiss Z2 LSM700 laser scanning confocal microscope at 1.5 µm/step using a 25X objective.

### 2.5. Commissure Analysis

Commissures of GABAergic (VD + DD GABAergic neurons, labeled by P*unc-47*::GFP or P*unc-47*::mCherry) and cholinergic motor neurons (labeled by P*unc-17*::GFP or P*acr-2*::mCherry [36]) were primarily scored at 63X using Zeiss Axio Imager M1 or Nikon90i microscopes equipped with fluorescence illumination. In the wild-type and mutant animals alike, a commissure is generally composed of a single neurite, and occasionally two neurites that extend together dorsally. Since neurite number cannot be reliably established for neurites labelled with the same color, we scored the morphology of commissures and express our data in terms of commissures rather than neurites. Thus, our values may represent an underestimate of defects that exist in the *ina-1* mutants.

Commissures that initially extended from the ventral nerve cord and failed to fully extend to the dorsal nerve cord were scored as incomplete. In these studies, three classes of incomplete commissures were scored: (1) premature bifurcation (T-like shape), (2) premature turn (L-like shape), and (3) premature stop (I-like shape). The “T” and “L” phenotypes are considered axon guidance defects because they make a premature turn and then extend longitudinally, while the “I” phenotype is considered a neurite outgrowth defect because it terminates prematurely before reaching the dorsal nerve cord (without any turn). The percentage of incomplete commissures and the percentage of animals with at least one incomplete commissure was calculated for each neuronal class (e.g., ACh vs. GABA).

### 2.6. Worm Staging

To obtain newly hatched L1 worms, embryos were picked to separate 60 mm unseeded plates and allowed to hatch for 30 min at 25 °C. Newly hatched L1 larvae were then moved to freshly seeded plates, and subsequently prepared for confocal imaging.

### 2.7. Fluorescence Intensity Analysis

To analyze soma and neurite fluorescence intensity of INA-1::GFP, background fluorescence was first measured in a region devoid of fluorescence and subtracted from each image using ImageJ software (open source). For quantifying INA-1::GFP in the soma, a defined 2 µm region of interest within the soma of GABAergic or cholinergic neurons labeled by cell-specific reporters was measured. For quantifying neurite INA-1::GFP intensity, GFP fluorescence was measured within a 20 µm region of a single confocal slice of the ventral nerve cord where co-labeling with cell-specific GABAergic or cholinergic signals could be clearly distinguished. The same exposure time and z-settings were used for all comparisons of fluorescence intensity across genotypes.

### 2.8. Statistical Analysis

Statistical comparisons of the percentages of incomplete commissures and animals were made using Fisher’s Exact test, with Bonferroni correction for multiple comparisons when applicable [37]. Error bars indicate standard error of proportion values unless otherwise indicated. Statistical comparisons of fluorescence intensity and average number of commissures per animal were made using unpaired *t*-test.

## 3. Results

### 3.1. α Integrin/ina-1 Is Necessary for Commissural Patterning in a Neuron-Specific Manner

*ina-1(gm144)* encodes a missense mutation that produces a proline to leucine change in the INA-1 extracellular domain proximal to the transmembrane region, and is suggested to alter ligand affinity [14,28,38,39]. We focused our efforts on animals carrying this mutation because it is the more severe of available *ina-1* hypomorphic alleles [28,29]. To define the role of α integrin/*ina-1* in the development of *C. elegans* motor neurons, we examined neurite extension in L4 transgenic worms co-expressing mCherry and GFP reporters in GABAergic and cholinergic neurons, using P*unc-47* and P*unc-17* promoters, respectively. In the wild type, GABAergic and cholinergic commissures, labeled in red and green, respectively, are easily visualized extending from the ventral nerve cord to the dorsal nerve cord, where they subsequently branch anteriorly and posteriorly in the dorsal nerve cord (Figure 1A).

We investigated GABAergic and cholinergic commissural outgrowth in wild type and *ina-1(gm144)* backgrounds and scored instances where commissures bifurcated prematurely, turned prematurely, or otherwise failed to reach the dorsal nerve cord as incomplete (Figure 1B). We observed a significant increase in the percentage of incomplete GABAergic commissures in *ina-1(gm144)* animals compared with wild type (Figure 1C), suggesting α integrin/*ina-1* is important for GABAergic commissural patterning, as previously reported [28,29,30,31]. We also quantified the number of worms with at least one GABAergic commissural defect. Interestingly, 28% of *ina-1(gm144)* animals scored had at least one GABAergic commissural defect (Figure 1D), with only 2 of 121 animals scored having >1 incomplete commissures. These commissural defects likely reflect deficits of both guidance and outgrowth, as we noted instances of improper commissural turning and continued lateral growth (T or L-shaped commissures, Figure 1B bottom panel, left and middle), as well as premature commissure termination (Figure 1B bottom panel, right). We noticed that the distal portions of many of the incomplete commissures had a blebby appearance (Figure 1A), perhaps indicating an accumulation of cytoplasmic material, and the possibility that *ina-1* may help localize factors that affect membrane protrusions [14,40,41]. We also noted a significant reduction in the average number of GABAergic commissures in *ina-1(gm144)* mutants compared with wildtype (*ina-1*: 11.1 ± 0.3, wt: 14.9 ± 0.2, *p* < 0.001), suggesting that the *ina-1(gm144)* mutation may interfere with neurite initiation from the ventral nerve cord or lead to more variable expression of the reporter. Finally, we noted occasional defasciculation of the ventral nerve cord, changes in the handedness of commissures (e.g., DD2), and variable cell body placement, as described previously [28,29,42].

In contrast to GABAergic commissures, we did not observe a significant increase in the percentage of incomplete cholinergic commissures in *ina-1(gm144)* animals. Specifically, we found less than 1% of incomplete cholinergic commissures in *ina-1(gm144)* mutants (0.8%), compared with 0.4% in wild type (Figure 1C). Similarly, the percentage of *ina-1* mutant worms with incomplete cholinergic commissures was not significantly different from wild type (Figure 1D). These data suggest that aspects of integrin signaling affected by the *ina-1(gm144)* mutation are not essential for cholinergic commissural patterning. We also examined the longitudinal process of the cholinergic DVA neuron, and did not observe obvious defects in neurite extension through the ventral nerve cord of *ina-*1 mutants (Appendix A). Together, our findings indicate that the integrin signaling functions impacted by the *ina-1(gm144)* mutation are more centrally involved in GABAergic commissural patterning, compared with cholinergic, suggesting a neuron class-specific role for integrin in commissural guidance.

### 3.2. GABAergic Commissural Defects in ina-1 Mutants Are Rescued by Expression of Wild Type ina-1

To confirm that the increased incidence of GABAergic commissural defects is due to reduced *ina-1* function, we expressed a wild type *ina-1* cDNA under control of the native *ina-*1 promoter (~4 kb) in *ina-1(gm144)* mutants and assessed rescue of the commissural defects (Figure 2). We found that wild type *ina-1* expression reduces the number of incomplete GABAergic commissures to a level that is comparable to wild type. To test if *ina-1* functions in the nervous system to promote commissural development, we investigated whether neuronal *ina-1* expression is sufficient to rescue the commissural defects of *ina-1(gm144)* animals. We found that *ina-1* expression in neurons (using the pan-neuronal promoter, P*rgef-1*) rescued the GABAergic commissural defects in one of three transgenic lines (Appendix A). Specific expression of wild type *ina-1* in GABAergic (using GABAergic specific promoter, P*unc-47*, 4 lines) or cholinergic (using the cholinergic specific promoter, P*unc-17β*, 2 lines) neurons did not offer significant rescue (Appendix A). Together, these results suggest that combined expression in multiple neuronal classes or additional tissues may be required for complete rescue of GABAergic commissural defects. In addition, our cell-specific expression experiments may not wholly recapitulate the timing and/or levels of expression necessary for proper guidance and extension of GABAergic commissures.

### 3.3. α Integrin/INA-1 Is Expressed in Both GABAergic and Cholinergic Motor Neurons

To better understand how ina-1 regulates GABAergic commissure outgrowth and guidance, we examined the expression of an ina-1 transcriptional reporter (Pina-1::GFP) using the same 4 kb promoter region that provided rescue above. We evaluated Pina-1::GFP expression in a transgenic strain where GABAergic motor neurons were labeled by Punc-47::mCherry. We observed low levels of Pina-1::GFP expression in GABAergic motor neurons of both L1 and L4 animals (Figure 3A–C). At the L1 stage, where only DD GABAergic neurons are present, we noted dim fluorescence in DD cell bodies that was obscured by comparatively high levels of Pina-1::GFP epidermal expression (Figure 3A,B). At the L4 stage, somatic fluorescence in GABAergic motor neurons remained dim, but was more clearly discernible in comparison with L1, likely due to decreased Pina-1::GFP expression in the epidermis of L4 stage animals (Figure 3C). We noted consistent low levels of expression across both VD and DD GABAergic neurons. We also noted stronger labeling of additional ventral cord motor neurons that were not labeled by the Punc-47::mCherry reporter, suggesting expression of ina-1 in other motor neuron classes.

To address this possibility, we generated a transgenic strain co-expressing the P*ina-1*::GFP reporter with the cholinergic marker P*acr-2*::mCherry. Overall, we noted that P*ina-1*::GFP fluorescence was stronger in cholinergic motor neurons compared with GABAergic motor neurons, and was expressed in only a subset of cholinergic motor neurons at both L1 and L4 stages (Figure 3D,E). At the L4 stage, P*ina-1*::GFP fluorescence was most consistently observed labeling the B-class cholinergic motor neurons (Figure 3E).

To investigate the subcellular localization of α integrin/INA-1, we expressed a rescuing fluorescently tagged INA-1 (P*ina-1*::INA-1::GFP) [28] together with transcriptional reporters labeling either GABAergic or cholinergic motor neurons (P*unc-47*::mCherry or P*acr-2*::mCherry, respectively) (Figure 4). Similar to previous studies, we noted that INA-1::GFP is present in many tissues in both L1 and L4 stage animals, in particular in neurons of the head, tail, and ventral nerve cord [28]. We found that INA-1 is expressed at low levels in DD GABAergic motor neuron somas of L1 stage animals (Figure 4A,B), and INA-1::GFP fluorescence was occasionally visible within GABAergic DD commissures (Figure 4C). At the L4 stage, we also noted low levels of INA-1::GFP expression in GABAergic neuronal cell bodies (Figure 4D), consistent with our above expression analysis. Notably, we also observed clear INA-1::GFP fluorescence in neurites that overlapped with P*unc-47*::mCherry expression, indicating significant levels of integrin in GABAergic processes (Figure 4E,G). Together, these results demonstrate that INA-1 is expressed in GABAergic motor neurons and readily transported into GABAergic neurites.

As noted above for the P*ina-1*::GFP transcriptional reporter (Figure 3), we found that INA-1::GFP fluorescence in the ventral nerve cord is not limited to GABAergic motor neurons (Figure 4A,F). To define the identities of these additional neurons with α integrin/INA-1 expression, we examined co-expression of INA-1::GFP with the P*acr-2*::mCherry cholinergic marker. At the L1 stage, we found that INA-1::GFP is localized to a subset of cholinergic motor neurons (Figure 4H). In contrast, by the L4 stage, INA-1::GFP is present in virtually all cholinergic motor neurons (Figure 4I). As noted for the P*ina-1*::GFP transcriptional reporter, average INA-1::GFP fluorescence intensity in cholinergic cell bodies was brighter in comparison to GABAergic cell bodies but is somewhat variable across cholinergic motor neurons, perhaps pointing to cell-type-specific regulation of integrin (Figure 4I,F). Notably however, neurites labeled by the cholinergic marker showed less robust INA-1::GFP fluorescence compared with GABAergic processes, suggesting less efficient distribution of integrin to the processes of cholinergic neurons (Figure 4G,J). A preferential localization of α integrin/INA-1 to GABAergic neurites is consistent with our above findings that INA-1 is important for GABAergic commissural patterning, and potentially points toward a site of action in GABAergic neurites. Further, the lack of cholinergic commissural defects in *ina-1* mutants suggests that INA-1 in cholinergic neurons may have functions independent of axon pathfinding, or there may be redundant mechanisms in cholinergic neurons to guide commissural outgrowth.

### 3.4. α Integrin/ina-1 Is Necessary for Embryonic and Post-Embryonic GABAergic Commissural Patterning

GABAergic motor neurons fall into two classes, DD and VD, based on the directionality of their synaptic innervation (dorsal D versus ventral D) and the timing of their birth (embryonic (DD) or post-embryonic (VD) born). Our observation that only one or two GABAergic commissures in each animal are affected by mutation of *ina-1* (Figure 1) raises the possibility that INA-1 may act preferentially to guide the extension of either DD or VD neurites. To address this question, we expressed the DD-specific P*flp-13*::mCherry transcriptional reporter in combination with the GABAergic P*unc-47*::GFP reporter in *ina-1(gm144)* mutants (Figure 5). In this doubly-labeled strain, DD neurons are identifiable by combined expression of red and green fluorescence, while VD neurons are solely labeled with green fluorescence. The percentage of incomplete commissures in *ina-1* mutants was not significantly different for VD versus DD neurons, indicating involvement of integrin in guiding neurite extension from both embryonic and post-embryonic born GABAergic neurons (Figure 5D). We found that 18% of worms had at least one defective DD commissure, while 16% of worms had at least one defective VD commissure. To account for the larger number of VD commissures in each animal compared with DD commissures, we normalized for the number scored in each class. This analysis also did not reveal a significant difference in the number of incomplete commissures across DD and VD neurons. These findings support involvement of integrin in the commissural patterning of both embryonic and post-embryonic born GABAergic motor neurons.

## 4. Discussion

Gaining a precise understanding of the spatiotemporal requirements for integrin signaling in directing neurite outgrowth and guidance is key for understanding molecular control of nervous system development. Here we took advantage of the simplified genetics offered by the *C. elegans* system to address questions about neuron-specific involvement of integrin signaling in neurite outgrowth and guidance. Specifically, we investigated the development of motor neuron commissural neurites in *C. elegans* strains carrying a hypomorphic mutation in the α integrin subunit *ina-1*. In wild type, both cholinergic and GABAergic neurons extend commissural projections from the ventral nerve cord to the dorsal nerve cord, and we found that *ina-1* is expressed by both motor neuron classes. Surprisingly however, mutation of *ina-1* preferentially impacts the development of GABAergic commissures. Our analysis revealed defects in both GABAergic commissural outgrowth and guidance, as indicated by occurrences of short, prematurely terminated commissures, as well as T- and L-shaped commissures. The defects in GABAergic commissures were variably rescued by specific expression of wild type *ina-1* in the nervous system, but not by specific *ina-1* expression in either GABAergic or cholinergic neurons. These results may suggest that combined expression in GABAergic and cholinergic ventral cord motor neurons is required for complete rescue, although we cannot exclude a contribution from other tissues such as epidermis. GFP-tagged INA-1 showed more robust localization to GABAergic neurites compared with cholinergic neurites, perhaps reflecting participation of this neurite-localized pool of integrin in GABAergic commissural patterning. Our analysis of *ina-1* in this simplified system points to differential involvement of α integrin/*ina-1* across structurally similar motor neuron classes.

We conducted an in-depth investigation of *ina-1* expression and localization in ventral cord motor neurons using transcriptional and translational reporters. In light of the fact that commissural defects in *ina-1* mutants were limited to GABAergic motor neurons, we were surprised to find comparatively higher levels of *ina-1* expression in cholinergic motor neuron somas using both methodologies. It is interesting to note that we observed significant differences across our transcriptional and translational reporter experiments. In general, the translational reporter showed more broad expression in the ventral nerve cord. For L4 stage cholinergic motor neurons, the INA-1::GFP translational reporter was detectable in essentially all cholinergic motor neurons labeled by the P*acr-2*::mCherry reporter. In contrast, expression of the transcriptional reporter was restricted to a subset of cholinergic motor neurons at both the L1 and L4 stages. These findings may point towards regulation of *ina-1* expression by intronic elements that are not present in the transcriptional reporter. Alternatively, our findings may suggest that INA-1::GFP protein stability is differentially regulated across motor neuron classes. For both motor neuron classes, we observed significant levels of INA-1::GFP in cell bodies at the L4 stage. This may in part reflect INA-1::GFP accumulation in the endoplasmic reticulum or arise as a consequence of overexpression [43,44]. INA-1::GFP fluorescence in GABAergic cell bodies was generally weaker than observed for cholinergic cell bodies, and we noted more prominent localization of INA-1::GFP to GABAergic neurites, indicating that INA-1::GFP was effectively transported and localized to GABAergic processes.

INA-1 is highly expressed in cholinergic neurons and these neurons extend commissures in a manner similar to GABAergic motor neurons; however, we did not observe obvious defects in cholinergic commissural outgrowth or guidance. Our analyses employed a hypomorphic allele that is predicted to alter ligand affinity and modify integrin function [28,38,39]. One interesting possibility is that the *gm144* allele interferes with specific aspects of integrin function, such as recruitment of key integrin effectors [14,40,41], that are solely required for GABAergic commissural development. α integrin/*ina-1* may also serve other functional roles in cholinergic motor neurons, such as in cell body positioning or fasciculation. This notion is consistent with our finding that INA-1::GFP fluorescence was most strongly visible in the cell bodies of cholinergic motor neurons, perhaps suggesting a primarily somatic site of action. Alternatively, the lack of an effect of the *ina-1(gm144)* mutation on cholinergic commissural patterning may suggest redundant mechanisms for directing this process in cholinergic motor neurons, in the form of either parallel molecular pathways for guidance or alternate integrin-based mechanisms. Prior expression studies have reported nervous system expression of the only other *C. elegans* α integrin, *pat-2* [45,46]. One possibility is that PAT-2 integrin acts redundantly with INA-1 in cholinergic motor neurons.

In principle, the incomplete GABAergic commissures in *ina-1(gm144)* animals could arise from impaired integrin function in the migrating neurites themselves or in neighboring tissues, such as the epidermis upon which the neurites extend. Our rescue experiments provide evidence that neuronal integrin expression contributes GABAergic commissural outgrowth and guidance, although we observed variability in the degree of rescue provided by neuronal expression of wild type *ina-1*. Our findings suggest a neuronal site of action, although contributions from multiple neuronal classes or other tissues may be required to fully restore GABAergic commissural development. We note that a prior study demonstrated GABAergic expression of wild type *ina-1* partially reverses aldicarb hypersensitivity in *ina-1(gm144)* mutants, providing evidence for cell autonomous *ina-1* function in GABAergic neurons in some contexts [47]. The defects in GABAergic commissural patterning that we observed in our work were often accompanied by gaps and thinning of the dorsal nerve cord, indicative of a failure to appropriately innervate dorsal musculature. This disruption of connectivity may contribute to mildly impaired motility in *ina-1(gm144)* mutants. Consistent with this, we noted that motility was improved by expression of wild type *ina-1* (not shown). Beyond the readily apparent defects in GABAergic neuron morphological features described here, it will be interesting in future studies to investigate additional potential roles for integrin in the establishment of motor circuit connectivity, for example in the development and/or placement of synaptic specializations.

## Figures and Tables

**Figure 1 jdb-07-00017-f001:**
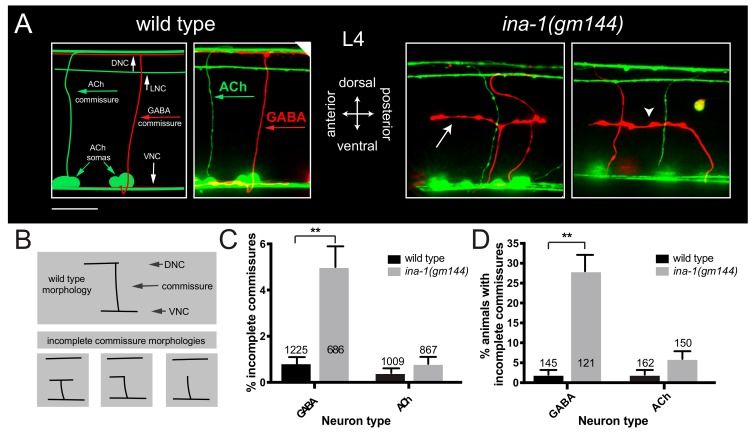
Commissural defects of *ina-1(gm144)* mutants. (**A**) Schematic (far left) and representative confocal image of wild type L4 GABAergic (GABA, red) and cholinergic (ACh, green) motor neuron somas, commissures, ventral nerve cord (VNC), dorsal nerve cord (DNC), and lateral nerve cord (LNC). Right: confocal images of GABAergic and cholinergic commissures in *ina-1(gm144)* L4 animals. Some GABAergic commissures prematurely bifurcate (arrow) and others prematurely turn prior to reaching the dorsal nerve cord (arrowhead). Scale bar, 10 µm. (**B**) Diagram of wild type commissural patterning and commissural defects scored. Wild type commissures fully extend between the ventral nerve cord and the dorsal nerve cord. In some *ina-1(gm144)* animals, commissures fail to reach the dorsal nerve cord, bifurcating prematurely (bottom left panel), turning inappropriately (bottom middle panel) or terminating prematurely (bottom right panel). These phenotypes were collectively scored as incomplete commissures. (**C**) Quantification of the percentage of incomplete commissures in wild type and *ina-1(gm144)* animals. The percentage of incomplete GABAergic, but not cholinergic, commissures in *ina-1(gm144)* animals is significantly increased compared to wild type. Fisher’s exact test, ** *p* < 0.0002. (**D**) The percentage of animals with incomplete GABAergic commissures (1 or more commissure defects) is significantly higher in *ina-1(gm144)* animals compared with wild type. Fisher’s exact test, ** *p* < 0.0002. Error bars are standard error of the proportion.

**Figure 2 jdb-07-00017-f002:**
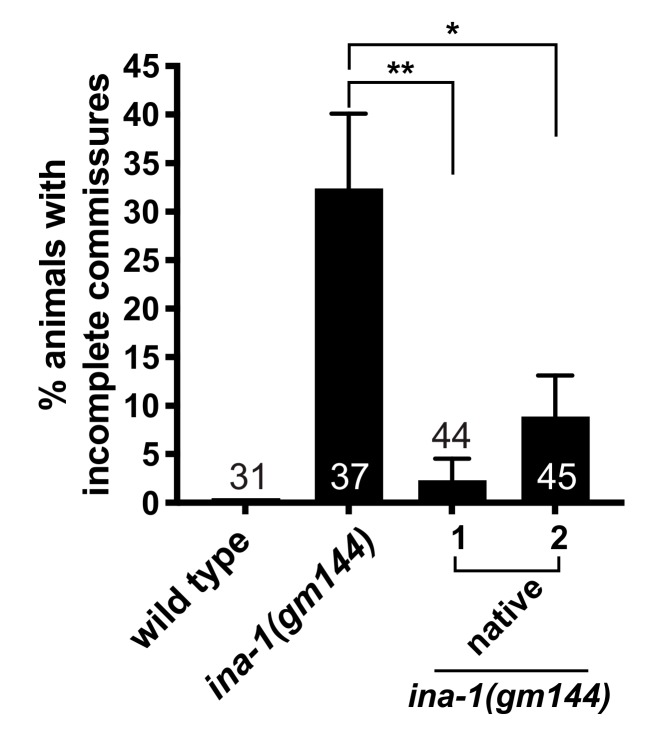
Expression of wild type α integrin/*ina-1* in *ina-1(gm144)* mutants rescues GABAergic commissural outgrowth and guidance. Commissural defects were assessed in L4 stage *ina-1(gm144)* mutants expressing wild type *ina-1*. Native refers to expression of *ina-1* cDNA using ~4.0 kb of regulatory sequence upstream of the *ina-1* start. * *p* < 0.03, ** *p* < 0.001, Fisher’s exact test with correction for multiple comparisons. Error bars are standard error of the proportion.

**Figure 3 jdb-07-00017-f003:**
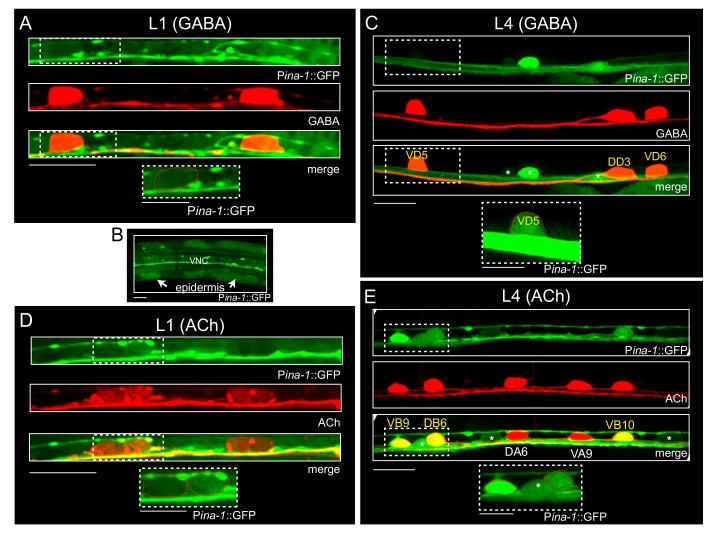
*ina-1* is expressed in cholinergic and GABAergic motor neurons. (**A**–**C**) Confocal images of the ventral nerve cord of animals expressing a transcriptional *ina-1* reporter (P*ina-1*::GFP, green) and GABAergic neuron marker (P*unc-47*::mCherry, red). (**A**) P*ina-1*::GFP is weakly expressed in DD GABAergic motor neurons during the L1 stage. Scale bar, 10 µm. Inset, magnified image of *ina-1* expression in DD GABAergic neuron. Scale bar, 5 µm. (**B**) High levels of epidermal expression during the L1 stage complicate visualization of neuronal fluorescence. Scale bar, 5 µm. (**C**) P*ina-1*::GFP is weakly expressed in GABAergic somas at the L4 stage (dashed box). P*ina-1*::GFP expression is also visible in other ventral nerve cord motor neurons (indicated by asterisks). Scale bar, 10 µm. Inset, high magnification image of P*ina-1*::GFP expression in GABAergic VD5 motor neuron soma (red outline). Intensity was increased to aid in visualization. Scale bar, 5 µm (**D**–**E**). Confocal images of the ventral nerve cord in animals expressing P*ina-1*::GFP together with the P*acr-2*::mCherry cholinergic motor neuron marker (ACh, red). (**D**) P*ina-1*::GFP is expressed in a subset of cholinergic motor neurons at the L1 stage (dashed box). Scale bar, 10 µm. Inset, magnified image of P*ina-1*::GFP expression in L1 stage cholinergic motor neuron somas (red outline). Scale bar, 5 µm. (**E**) P*ina-1*::GFP is strongly expressed in a subset of cholinergic motor neurons (likely B-type cholinergic neurons) at the L4 stage. Asterisks denote P*ina-1*::GFP expression in additional ventral nerve cord neurons not labeled by P*acr-2*::mCherry. Some of these may be AS neurons that are not efficiently labeled by this marker. Scale bar, 10 µm. Inset, magnified image of P*ina-1*::GFP expression in L4 stage cholinergic motor neuron somas (red outline). Scale bar, 5 µm.

**Figure 4 jdb-07-00017-f004:**
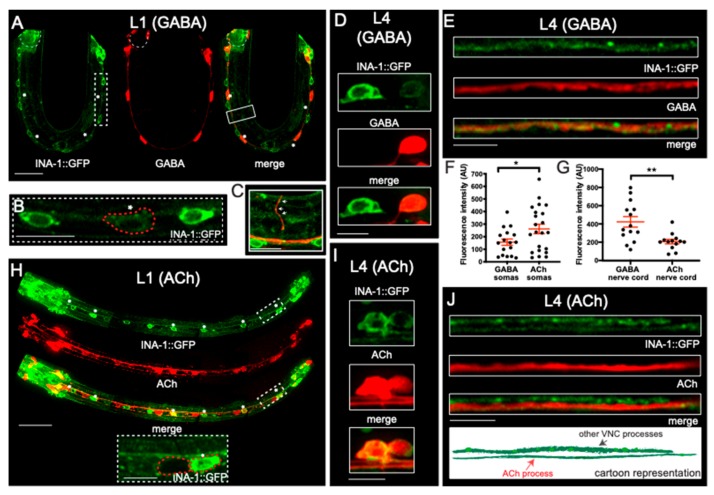
INA-1 localizes to the somas and neurites of GABAergic and cholinergic neurons. (**A**–**E**) Confocal images of transgenic strains expressing INA-1::GFP with the GABAergic motor neuron marker P*unc-47*::mCherry (GABA, red). (**A**) In newly hatched L1s, INA-1::GFP is weakly localized to the cell bodies of DD neurons (dashed box, magnified in **B**). Asterisks indicate GABAergic neurons expressing INA-1::GFP. Dashed outlines indicate the pharynx. Solid rectangle indicates commissure, magnified in C. Note that INA-1::GFP is also visible in additional motor neuron classes. Scale bar, 10 µm. (**B**) Magnified view of INA-1::GFP expression in L1 stage ventral cord motor neurons. GABAergic motor neuron outlined in red. Scale bar, 10 µm. (**C**) Magnified view of GABAergic commissure (dashed white box in A). INA-1::GFP localization is detectable in some GABAergic commissures (white arrows). Scale bar, 5 µm. During L4, INA-1::GFP localizes to GABAergic somas (**D**,**F**) and neurites (**E**,**G**) (single confocal section) within the ventral nerve cord. Scale bars, 5 µm. (**F**,**G**) Quantification of fluorescence intensity in GABAergic and cholinergic ventral nerve cord motor neuron somas (**F**) and processes (**G**). (**F**) INA-1::GFP signal is significantly higher in cholinergic somas compared to GABAergic. GABAergic somas (*n* = 17), cholinergic somas (*n* = 22), unpaired *t*-test * *p* ≤ 0.05. (**G**) INA-1::GFP signal is significantly higher in GABAergic processes compared to cholinergic. GABAergic (*n* = 14), ACh (*n* = 13), unpaired *t*-test, ** *p* ≤ 0.01. (**H**–**J**) Confocal images of transgenic strains expressing INA-1::GFP together with the cholinergic motor neuron marker P*acr-2::mCherry*. (**H**) In L1, INA-1::GFP is present in a subset of cholinergic somas (asterisks). Dashed box region is magnified in inset. Scale bar, 20 µm. Inset, magnified view of L1 ventral nerve cord showing INA-1::GFP localization in a subset of cholinergic neurons (red outlines). Scale bar, 5 µm. (**I**) INA-1::GFP is expressed at varying levels in most cholinergic somas at the L4 stage. Scale bar, 5 µm. (**J**) Single confocal section of ventral nerve cord in L4 animals. INA-1::GFP is diffusely localized in cholinergic neurites but stronger INA-1::GFP fluorescence is evident in an adjacent subset of neurites not labeled by the P*acr-2::mCherry* cholinergic motor neuron reporter. Based on 4G, these are likely to be GABAergic neurites. Scale bar, 5 µm. Bottom, cartoon representation of confocal images (VNC = ventral nerve cord).

**Figure 5 jdb-07-00017-f005:**
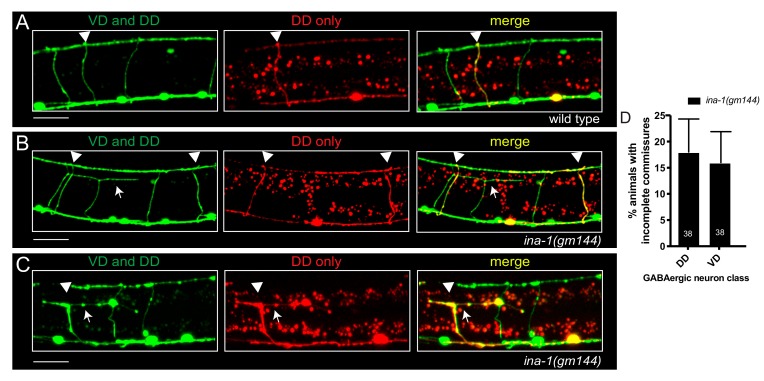
*ina-1* is important for commissural patterning of both embryonic and post-embryonic born GABAergic neurons. (**A**–**C**) GABAergic commissures were visualized in animals expressing P*unc-47*::GFP (DD and VD neurons, green) together with P*flp-13*::mCherry (DD neurons, red). Commissures with both red and green fluorescence (yellow) are DD neurons, while VD commissures are labeled solely by green fluorescence. (**A**) Wild type commissures of each neuron class fully extend from the ventral to the dorsal nerve cord. Misguided VD (**B**) and DD (**C**) commissures are observed in *ina-1(gm144)* mutants. White arrows indicate prematurely bifurcating “T” shaped commissures. White arrowheads indicate DD commissures. Scale bar, 20 µm. Note that defects in GABAergic commissures often produce gaps or thinning within the dorsal nerve cord (**B**,**C**). (**D**) VD and DD commissures are similarly affected by mutation of *ina-1*. Error bars indicate standard error of the proportion.

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
