# Peer review of "Integrins Have Cell-Type-Specific Roles in the Development of Motor Neuron Connectivity"

_jdb, 2019, doi:10.3390/jdb7030017_

Round 1

Reviewer 1 Report

This paper presents studies of the role of a C. elegans integrin, ina-1, in the morphology of body commissural neuronal processes. The authors suggest that while ina-1 is expressed in both cholinergic and GABAergic neurons, it has preferential effects on two subclasses of GABAergic neurons. The authors provide evidence that ina-1 function is required in neurons.

While many proteins have been implicated in process guidance in the nervous system, a complete explanation for how most neuronal processes are guided is lacking. The work here adds another gene into the mix, which may be useful eventually in understanding patterning rules in the nervous system. The paper does not delve into mechanism of action, and essentially just reports on the ina-1 mutant phenotype, and expression. ina-1 was previously implicated in axon guidance, so the novelty of the report here mainly lies in characterizing the roles in the specific neuron types considered.

While some of the results are strong, and conclusions warranted, there are some issues that should be addressed:

1. The authors suggest ina-1 functions in neurons, showing rescue of defects with ina-1 cDNA expressed in 1/2 transgenic lines. However, it appears that for muscle and epidermis only one line was tested. It remains possible that the one neuronal rescuing line is an experimental fluke. The authors need to look at additional lines for all tissues to draw a stronger conclusion.

2. Since ina-1 stains mainly processes, it’s hard to know whether the expression is in the process of GABAergic cells or in adjacent processes. Indeed in the images the fine details in morphology of the cell reporter and the ina-1 reporter do not always overlap.

3. To show that ina-1 functions in GABAergic neurons, it seem very important to perform the cDNA rescue experiment in these neurons.

4. The authors suggest the lack of effects of the ina-1 mutation on cholinergic neurons could be due to failure of the protein to localize to processes, even though overall expression is higher than in GABAergic neurons. What then would be the purpose of ina-1 in the cholinergic cells? Is it possible that some of the effects of ina-1 on GABAergic neurons comes non-autonomously from the cholinergic neurons?

 In sum, this is a preliminary characterization of the role of ina-1 in GABAergic neuron morphogenesis. While most conclusions seem reasonable, the authors do not explore mechanism of action, or provide definitive evidence for ina-1 site of action.

Reviewer 2 Report

In this manuscript by Oliver et al, the authors investigate the role of alpha- integrin INA-1 in proper extension and guidance of C. elegans motor neuron (MN) commissures. They use a hypomorphic point mutant of ina-1 (gm144) in all their analyses, because the null mutant is early lethal. Using this allele, they show defects in ina-1 mutant commissure formation by GABAergic MNs, but not cholinergic MNs, despite expression of ina-1 in both types of neurons. This quantification and specificity is a new finding, although the commisure defect in GABA neurons was first shown qualitatively by Poinat et al. The authors carefully analyse ina-1 expression in these MN’s, using transgenes with GFP. They show both ina-1 promoter expression and GFP-tagged INA-1 protein localization of alpha integrin in both GABAergic and cholinergic neurons. Interestingly, ina-1(gm144) affects both embryonic and post-embryonic GABAergic MN commissures. The authors also address the important question of where this integrin is required for proper axon guidance: in the neurons, muscle, or epidermis (possible substrate cells). However, the result there was not as conclusive as claimed (see below).  The images are of good quality, and the careful demonstration that there is weak but consistent expression in GABA MNs is taken, but a few images need to be made larger or include more indicators. The manuscript was written well overall, but a few other ideas about cellular mechanism would make the discussion more interesting.     

Major points: 

1.)  This allele is a missense mutation changing a proline to leucine, and no evidence is provided (and I couldn’t find in the literature) that the INA-1 protein is reduced. Therefore, the multiple places where it’s stated that “ina-1 is required for X..” are overstatements. It is not clear exactly how much or what specific function of ina-1 is lost in this mutant. They say this point mutation “is predicted to interfere with integrin activation”, citing Ref 28. But Ref 28 makes a guess based on citing a paper done with a different species alpha-integrin (which should be cited also), so the reality is that this allele could represent some very specific partial loss of function of ina-1 dependent on that proline, rather than a simple reduction of all ina-1 functions. It is important to step back these “required” statements. It may be that the proline is required, or a sub-function mediated by that proline is required. If gm144 was shown by immunoblot to cause reduced overall INA-1 protein levels, or, if there were another hypomorph allele with a different mutation, that were also tested and had similar phenotypes as gm144, then the “requirement” statements would be more justified.

2.)  Related to the possible reason for the specificity of the phenotype shown here, it would add to the interest in the Discussion to speculate, based on known data on ina-1 or other alpha integrins, as to possible reasons for the specificity of the phenotype for GABA MN’s and not cholinergics. Is there compensation by pat-2 or ten-1? Does the proline mediate some function that only the GABA neurons need? what aspect of interaction or activation might be disabled by this point mutation. The Sherwood lab has a few very interesting papers showing ina-1 may help localize factors to the plasma membrane that affect membrane protrusions. Perhaps the blebbing noted in Fig 1A mutant axons could be related to this?

3.)  Figure 2 addresses the important question of cell-autonomy, whether ina-1 is required in the axons or in other cell types by using different promoters to try for cell-type specific rescue of the commissure phenotype. However, the conclusion that ina-1 is required in neurons and not in muscle or epidermis is not convincing. There are two pan-neuronal lines, and one rescues and one doesn’t. There is only one muscle line and one epidermis line. If there were two of each, would it be 50/50 for all? Testing three (non- integrated) lines for each rescue plasmid is the standard, to be able state a reasonably confident conclusion about whether it does or does not rescue. If three lines were tested for each plasmid, then the mean percentages from the three different lines could be compared using ANOVA, which is the proper statistical method for multiple comparison testing. Finally, expressing percentages out of 29 is iffy. If additional lines cannot be made, then the experiment should be taken out or the conclusion from it withdrawn. Although not necessary, an even better experiment would be to use the GABAergic specific promoter rather than pan-neuronal.

4.)  In Figure 4 E and H, the argument that there is a difference in protein levels of INA-1::GFP between GABAergic and cholinergic neurites was not supported by the images (they look to be the same intensity) and was not quantified, and therefore not convincing. It is hard to determine if there is more or less INA-1::GFP between these two sets of pictures. It would be important to quantify the amount of axonal GFP using a measure of fluorescent intensity like a line scan and compare statistically between GABAergic and cholinergic neurons. (And the images would need to be collected with similar exposure times and over multiple GFP lines, averaged.) Alternatively the conclusion should be walked back in the results and discussion.  

Minor points: 

Specific comments:  For this manuscript to be accepted the authors would need to address the points below:

5.)  Figure 1: The ratios in C and D between animal number and commissure number, being different for wt versus mutant, would suggest there are fewer commissures in the ina-1 mutants. Do the ina-1 mutants actually have fewer commissures (secondary to neuron migration defect perhaps?)? If they don’t, why were fewer commissures examined in the mutants? The authors should make clear: How many axons per commissure, and how many commissures per animal in the wild type? Regarding the low penetrance of the phenotype, is it surprising that only one commissure has a defect per animal? (It is stated that only 2/121 have more than one commissure appearing incomplete.) Does one commissure being incomplete represent only one axon being miswired, or a group of axons?

6.)  Clarify labelling in Figure 1A to indicate anterior and posterior directions. Also it shoud be Ach somas not soma (or somata) for multiple cell bodies. 

7.)  Figure1A right side: please use some marker like an asterisk to point out the fasciculations. This is an interesting phenotype. Can the authors quantify the % that had these fasciculations, were these fasciculations only on incomplete commissures or also on the commissures that successfully crossed?

8.)  It would be helpful to have the developmental stage (L4) being analysed labeled somewhere on the figure, in addition to in the legend. Perhaps on the very left side of A.

9.)  Figure1B: Breaking down the incomplete commissures from 1C into percentages for these three subtypes of incomplete commissures morphologies would be interesting. What is the frequency of each type of incomplete commissure? If one type is prevalent, that could inform cellular function of ina-1. The “T and L shape” description used in the discussion could be used when they are first described, to keep the wording consistent. Additionally, in the methods, a clearer description of how these categories were determined would be informative.   

10.)                 Although this is the common method for expression studies in C. elegans, it should be acknowledge for both Figure 3 and 4 that the transgenic data may not represent endogenous RNA or protein expression. For Figure 3 that the images show the level of GFP being transcribed by the promoter – so it is showing promoter activity. For the GFP protein tag of INA-1, this represents overexpression not endogenous protein, so the GFP protein may go to locations that INA-1 normally would not.

11.)                 Figure 3 and 4:  minor figure changes needed:

a.    Dashed boxes in 3A and 3D are not visible 

b.    arrows in 3B are too small

c.    In Figure 3, it would help the reader to add the word “GABA” or “ACh” next to the word L1 or L4 at the top of each quadrant, so it is more clear which type of neuron is being examined with the mCherry plasmid labeling in each part of the figure. 

d.    4C is too small to see – making it bigger is necessary

12.)                 Figure 5/5D needs a callout in the main text, somewhere around lines 295-300. 5D graph is too small to see, and the graph with normalization for number scored should be included. 

13.)                 For Figure S1, how many worms were examined to make this conclusion? Please state the n in text or legend.

14.)                 For figure S2, what is the significance of this? Does the puncate staining indicate varicosities? Synapses? Was this observed in many samples? Does this suggest something about cellular function of INA-1? Please explain briefly in text or leave it out. 

15.)                 Line 326 says, “short, prematurely terminated commissure.” The authors didn’t measure length of the commissure, therefore the use of the word short seems inappropriate because it implies that the axons have a defect in outgrowth. The authors and others have shown INA-1 is important for guidance but not axonal outgrowth. If axons are suspected to actually be shorter, could the authors measure the length of axons or bundle of axons? It would be interesting to know if the commissure grows a similar length just in the wrong direction. 

Round 2

Reviewer 1 Report

The authors have edited the original manuscript and performed some new experiments. Unfortunately, the new studies do not shed light on the site of action of ina-1. Since ina-1 was previously shown to act in guidance, the main novelty of the current manuscript is in describing the specific role in the specific neuron subclasses examined here. I worry that without the site of action, the paper may not make a clear cut advance. Is it possible to perform cell-specific RNAi, or to use cell-specific degradation of INA-1 protein (using the zif-1 system, for example) to determine which cells are relevant?

Reviewer 2 Report

The revisions made to the manuscript are satisfactory and I think the revised manuscript should be accepted.